# Iodide-enhanced palladium catalysis via formation of iodide-bridged binuclear palladium complex

Yuanfei Zhang [1,3], Zhe-Ning Chen [1,3], Xiaofeng Zhang[1,3], Xi Deng[1], Wei Zhuang[1✉] & Weiping Su [1,2✉]

The prevalence of metalloenzymes with multinuclear metal complexes in their active sites inspires chemists' interest in the development of multinuclear catalysts. Studies in this area commonly focus on binuclear catalysts containing either metal-metal bond or electronically discrete, conformationally advantageous metal centres connected by multidentate ligands, while in many multinuclear metalloenzymes the metal centres are bridged through μ2-ligands without a metal-metal bond. We report herein a μ2-iodide-bridged binuclear palladium catalyst which accelerates the C-H nitrosation/annulation reaction and significantly enhances its yield compared with palladium acetate catalyst. The superior activity of this binuclear palladium catalyst is attributed to the *trans* effect-relay through the iodide bridge from one palladium sphere to the other palladium sphere, which facilitates dissociation of the stable six-membered chelating ring in palladium intermediate and accelerates the catalytic cycle. Such a *trans* effect-relay represents a bimetallic cooperation mode and may open an avenue to design and develop multinuclear catalysts.

---

[1] State Key Laboratory of Structural Chemistry, Fujian Institute of Research on the Structure of Matter, Chinese Academy of Sciences, 155 Yangqiao Road West, Fuzhou 350002, PR China. [2] Center for Excellence in Molecular Synthesis, Chinese Academy of Sciences, Shanghai 200032, PR China. [3] These authors contributed equally: Yuanfei Zhang, Zhe-Ning Chen, Xiaofeng Zhang. ✉email: wzhuang@fjirsm.ac.cn; wpsu@fjirsm.ac.cn

The prevalence of the metalloenzymes containing binuclear or polynuclear transition metal complexes in their active sites has motivated the interest of chemists in the binuclear or polynuclear homogeneous catalysts for organic transformations[1–10]. A number of well-defined binuclear catalysts have been reported with unique activity and selectivity accessible through the cooperation between two metal centres[1,4–6,8–12]. Two types of bimetallic cooperation are commonly encountered in the catalytic processes of these binuclear catalysts: one is the simultaneous activation of two reaction partners by the binuclear catalysts in which multidentate ligands hold two metal centres in close proximity to create two electronically discrete, conformationally advantageous active sites for binding the corresponding substrate molecules[1,4,5,13,14], as illustrated by olefin polymerization[13] or enantioselective binuclear catalysts[5,14]; the other is that two metal centres of the metal–metal bond-containing binuclear catalyst synergistically participate in elemental redox steps in catalytic pathways[6,9,10,15–32], as exemplified by Pd(III) dimer-catalysed oxidative C–H functionalization[17–19] and dirhodium-catalysed reactions for carbene[20–22] or nitrene insertion[23] into C–H bond. In many of the metalloenzymes, on the other hand, metal centres at the active site are held together by μ2-bridging ligands, with the metal–metal bond absent, implicating that the catalysis therein may invoke different bimetallic or polymetallic cooperation in which μ2-bridging ligands likely play key roles. Disclosing the bimetallic or polymetallic cooperation mechanism for catalysis would provide platforms to rationally design the binuclear or polynuclear metal catalysts with the unique activity and selectivity. Due to the rapid equilibrium of the multinuclear complexes with other metal species in the catalytic conditions, however, it has been highly challenging to identify the active species and reveal the responsible cooperation mechanism in a definitive manner[7], which is the reason why the promising binuclear or multinuclear metal catalysts are relatively underdeveloped compared with the mononuclear metal catalysts that dominate in homogenous catalysis.

Here, we report an iodide-bridged binuclear palladium catalyst generated in situ from palladium acetate, azobenzene and tetra-n-butyl ammonium iodide (TBAI), which accelerates C–H nitrosation/annulation reaction and significantly enhances its yields compared with palladium acetate alone as a catalyst (Fig. 1). This binuclear palladium species, according to the kinetic studies, retains the integrity of its iodide-bridged binuclear core structure during the catalytic cycle. Computational studies further reveal that a strongly σ-donating η¹ phenyl ligand around one palladium centre of this binuclear complex is able to exert a trans effect[33], through the bridging iodide ligand, on the ligand at the other palladium centre and therefore labilize the coordination bond trans to this bridging iodide[34,35]. Consequently, two palladium centres within the binuclear cluster cooperatively decrease the activation barriers of dissociation of the chelating product fragment from catalyst and accelerate the whole catalytic cycle. The binuclear metal catalyst that features the trans effect-relay through bridging ligand may provide a solution to metal-catalysed efficient syntheses of the chelating compounds that often impede catalytic cycle.

## Results and discussion

**Discovery of beneficial effect of iodide.** Our interest in the binuclear palladium catalyst stemmed from a discovery during exploration of palladium-catalysed ortho C–H bond nitrosation reaction of azobenzenes. The development of this Pd-catalysed aryl C–H nitrosation was aimed at achieving a straightforward approach to nitrosoarenes that are a class of versatile synthetic intermediates utilized in a variety of transformations[36]. Importantly, the metal-catalyzed aryl C–H nitrosation method has a potential to get over substrate limitation, poor regioselectivity[37]

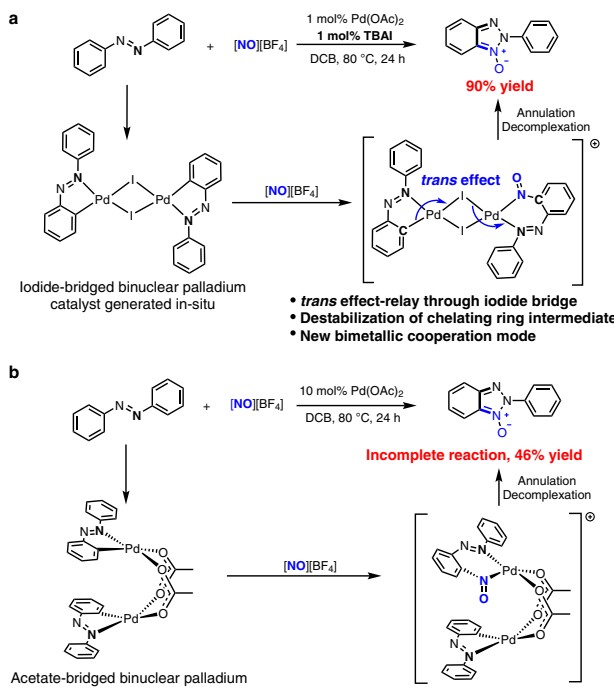

**Fig. 1 Palladium-catalysed C–H nitrosylation/annulation reaction of azobenzene with [NO][BF₄]. a** The equimolar combination of (n-Bu₄N)I with Pd(OAc)₂ leads to enhancement of palladium catalysis via formation of the iodide-bridged binuclear palladium catalyst that is able to destabilize six-membered chelating ring of reaction intermediate due to trans effect-relay through iodide bridge, and therefore facilitate the reaction. The trans effect-relay through iodide bridge represents a bimetallic cooperation mode for catalysis. **b** The reaction catalyzed by 10 mol% Pd(OAc)₂ alone affords lower yield than the reaction with combination of 1 mol% (n-Bu₄N)I with 1 mol% Pd(OAc)₂. The difficult in decomplexation of six-membered chelating ring of reaction intermediate retards the reaction catalysed by Pd(OAc)₂ alone and results in incomplete reaction.

and substrate pre-activation[36] problems encountered in the existing methods for syntheses of nitrosoarenes. To this end, we started our investigation by examining the reaction of azobenzene (**1a**) with two equivalents of nitrosonium tetrafluoroborate ([NO][BF₄]) (**2**) conducted in 1,2-dicholobenzene (DCB) of 1.5 mL at 80 °C for 24 h in the presence of 1 mol% Pd(OAc)₂ as a catalyst (Table 1). The initial reaction conditions enabled the desired ortho C–H nitrosation but generated 2H-benzotriazole N-oxide product (2-phenyl-2H-benzo[d][1,2,3]triazole 1-oxide **3a**) as a final product in only 7% yield via a C–H nitrosation/annulation sequence (entry 1 in Table 1).

The fact that 2H-benzotriazole heterocyclic N-oxides are the privileged structural motifs in biologically active compounds, pharmaceuticals, and functional materials[38–40] prompted us to identify the optimal conditions for this Pd-catalysed reaction of azobenzene with [NO][BF₄]. Increasing catalyst loading to 10 mol% offered a 46% yield of the desired **3a** with azobenzene recovered (entry 2 in Table 1). Screening a variety of additives revealed that 10 mol% tetra-n-butyl ammonium chloride (TBAC), in combination with 10 mol% Pd(OAc)₂, enhanced the yield of **3a** to 76% (entry 3 in Table 1). Interestingly, TBAI exhibited a more beneficial effect on the reaction than TBAC. 1 mol% TBAI allowed using 1 mol% Pd(OAc)₂ to achieve a 90% yield (entry 4 in Table 1). Considering that p-toluenesulfonic acid (TsOH) may facilitate cyclopalladation of azobenzene, 2 mol% TsOH was introduced into the reaction system with 1 mol% TBAI and 1 mol% Pd(OAc)₂, but still afforded a 90% yield (entry 5 in Table 1).

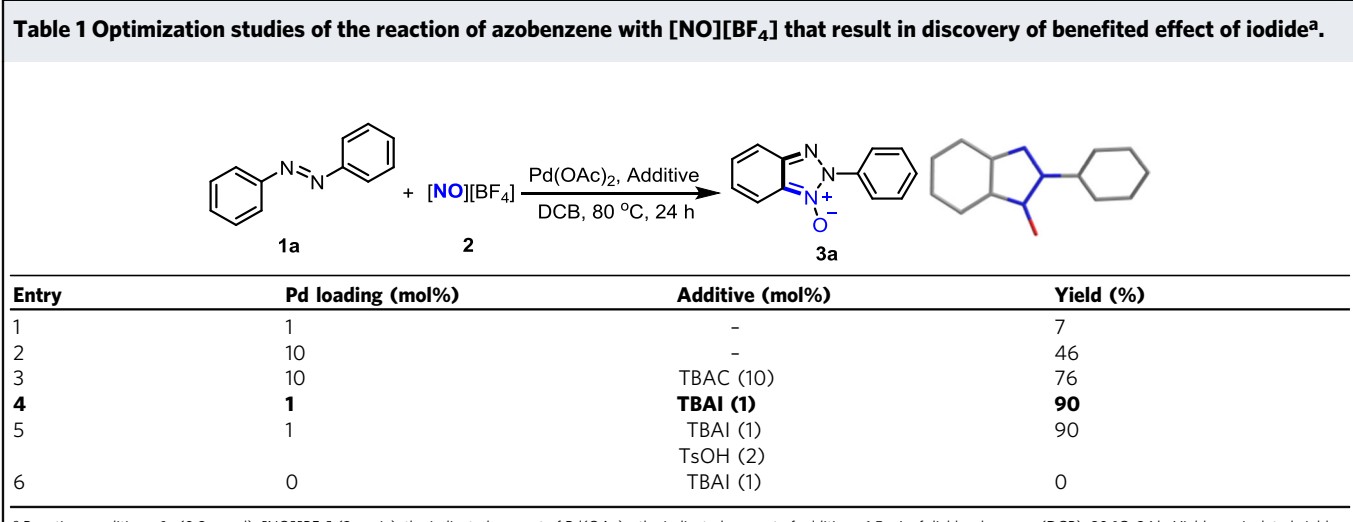

**Table 1 Optimization studies of the reaction of azobenzene with [NO][BF$_4$] that result in discovery of benefited effect of iodide[a].**

| Entry | Pd loading (mol%) | Additive (mol%) | Yield (%) |
|---|---|---|---|
| 1 | 1 | – | 7 |
| 2 | 10 | – | 46 |
| 3 | 10 | TBAC (10) | 76 |
| **4** | **1** | **TBAI (1)** | **90** |
| 5 | 1 | TBAI (1) TsOH (2) | 90 |
| 6 | 0 | TBAI (1) | 0 |

[a] Reaction conditions: **1a** (0.2 mmol), [NO][BF$_4$] (2 equiv), the indicated amount of Pd(OAc)$_2$, the indicated amount of additives, 1.5 mL of dichlorobenzene (DCB), 80 °C, 24 h. Yields are isolated yields.

Other protonic acids did not show beneficial effect on the reaction yield as well (see Supplementary Table 1). Control experiment showed that Pd(OAc)$_2$ was indispensable for this reaction (entry 6 in Table 1). The beneficial effect of iodide on the Pd-catalysed C–H nitrosation/annulation sequence is to some extent surprising because it is at variance of the previous findings that iodide anion tends to retard late transition metal-catalysed C–H functionalization reactions by weakening electrophilicity of transition metal ions[41,42]. Although halide anions including iodide enhance Pd-catalysed traditional cross-coupling reactions of aryl halides with nucleophilic organometallics[43], these Pd-catalysed traditional cross-coupling reactions via Pd(0)/Pd(II) cycle are distinctly different from the Pd-catalysed oxidative C–H nitrosation/annulation reaction that would begin with the electrophilic cyclopalladation of Pd(II) species and involves Pd(II)/Pd(III)[17–19] or Pd(II)/Pd(IV)[18,44,45] cycle.

**Identification of iodide-bridged binuclear palladium complex as a catalytically active intermediate**. Intrigued by this beneficial effect of iodide, we performed mechanistic investigation to identify its origin. Initially, we tried to isolate a palladium intermediate lying on catalytic cycle. Under the conditions mimicking the aforementioned Pd(OAc)$_2$/TBAI-catalysed reaction, the reaction of 0.2 mmol Pd(OAc)$_2$ with one equivalent of TBAI, 100 equivalents of azobenzene and 200 equivalents of [NO][BF$_4$] afforded an iodide-bridged binuclear palladium complex bearing cyclopalladated azobenzene ligand as dark red solid (**4a**) (Fig. 2a). Single-crystal X-ray diffraction analysis revealed that the molecule of **4a** comprises a twofold iodide-bridged binuclear palladium core with each palladium atom chelated by an azobenzene ligand through nitrogen and carbon atoms (Fig. 2a)[46], and square planar coordination environments around these two palladium atoms are identical and lie in the same plane because they are related by centrosymmetry. Within **4a**, the I–Pd bond *trans* to Pd–C bond is 0.12 Å longer than the other I–Pd bond between the iodide atom in question and the other palladium centre owing to strong *trans* influence of phenyl ligand, leading to unsymmetrical iodide bridges. Because the poor solubility of crystalline **4a** in 1,2-dichlorobenzene impeded the investigations of reactivity of **4a**, we turned our attention to synthesis of highly soluble congeners of **4a**. The treatment of Pd(OAc)$_2$ with one equivalent of TBAI and 12 equivalents of 4,4′-di-n-butyl-azobenzene (**1b**) in DCB at 80 °C for 12 h afforded a soluble analogue of **4a**, an iodide-bridged binuclear palladium complex of cyclopalladated 4,4′-di-n-butyl-azobenzene ligand (**4b**) in 65% yield with

concomitant formation of [Pd$_2$I$_6$](n-Bu$_4$N)$_2$[47] in 11% yield (Fig. 2b). The cyclopalladated azobenzene ligands in **4a** and **4b** indicate that Pd-introduced C–H bond cleavage can occur in the presence of iodide anion, though iodide anion was thought to tend to form stable Pd–I bond and thus weaken the Pd electrophilicity that is required for C–H metallation[41].

Using soluble **4b** as a model compound, reactivity and kinetics of the iodide-bridged binuclear palladium complexes were then examined to identify their roles in the catalytic process. The reaction of **4b** with 3 equivalents of [NO][BF$_4$] at room temperature for 24 h showed that cyclopalladated 4,4′-di-n-butyl-azobenzene ligand in **4b** underwent nitrosation and subsequent annulation to form 2H-benzotriazole N-oxide derivative bearing two n-butyl groups (**3b**) in 38% yield (Fig. 2c), which raised the possibility that such an iodide-bridged binuclear palladium complex was a reaction intermediate in the catalysis cycle. Furthermore, 1.5 mol% **4b**, together with 3 mol% tetra-n-butyl ammonium acetate (n-Bu$_4$NOAc) as an additive, served as a catalyst to effect the nitrosylation/annulation reactions of both 4,4′-di-n-butyl-azobenzene (**1b**) and 4,4′-dimethyl-azobenzene (**1c**) in the yields (Fig. 2d) comparable with those obtained with the combination of 3 mol% Pd(OAc)$_2$ and 3 mol% TBAI (see below). The use of 3 mol% n-Bu$_4$NOAc as an additive in these **4b**-catalysed reactions is because the reaction of azobenzene catalysed by Pd(OAc)$_2$ and TBAI might generate in situ the iodide-bridged binuclear palladium complex along with 2 equivalents of n-Bu$_4$NOAc and 2 equivalents of acetic acid. In the **4b**-catalysed transformation of azobenzenes to the corresponding 2H-benzotriazole N-oxide products, 1.5 mol% **4b** provided slightly higher initial reaction rate compared with the combination of 3 mol% Pd(OAc)$_2$, 3 mol% TBAI and 6 mol% TsOH, indicating that this binuclear palladium complex **4b** is a kinetically competent catalyst (Fig. 2e). In the experiments determining the initial rate for reaction with 3 mol% Pd(OAc)$_2$ and 3 mol% TBAI, 6 mol% TsOH was introduced as an additive into the reaction system to neutralize OAc⁻ anion from Pd(OAc)$_2$ and rule out the influence of OAc⁻ when comparing with the reaction with **4b**. The initial rates of the nitrosylation/annulation of **1c** exhibited a first-order dependence on the concentration of **4b** (Fig. 2f), supporting that the iodide-bridged binuclear palladium complex **4b** and its congeners are catalytically active species, and retain integrity of its [Pd$_2$I$_2$] core structure during catalysis. The first-order dependence of the initial reaction rate on the concentration of **4b** may also account for the observation that the reaction catalysed by 1.5 mol% **4b** was

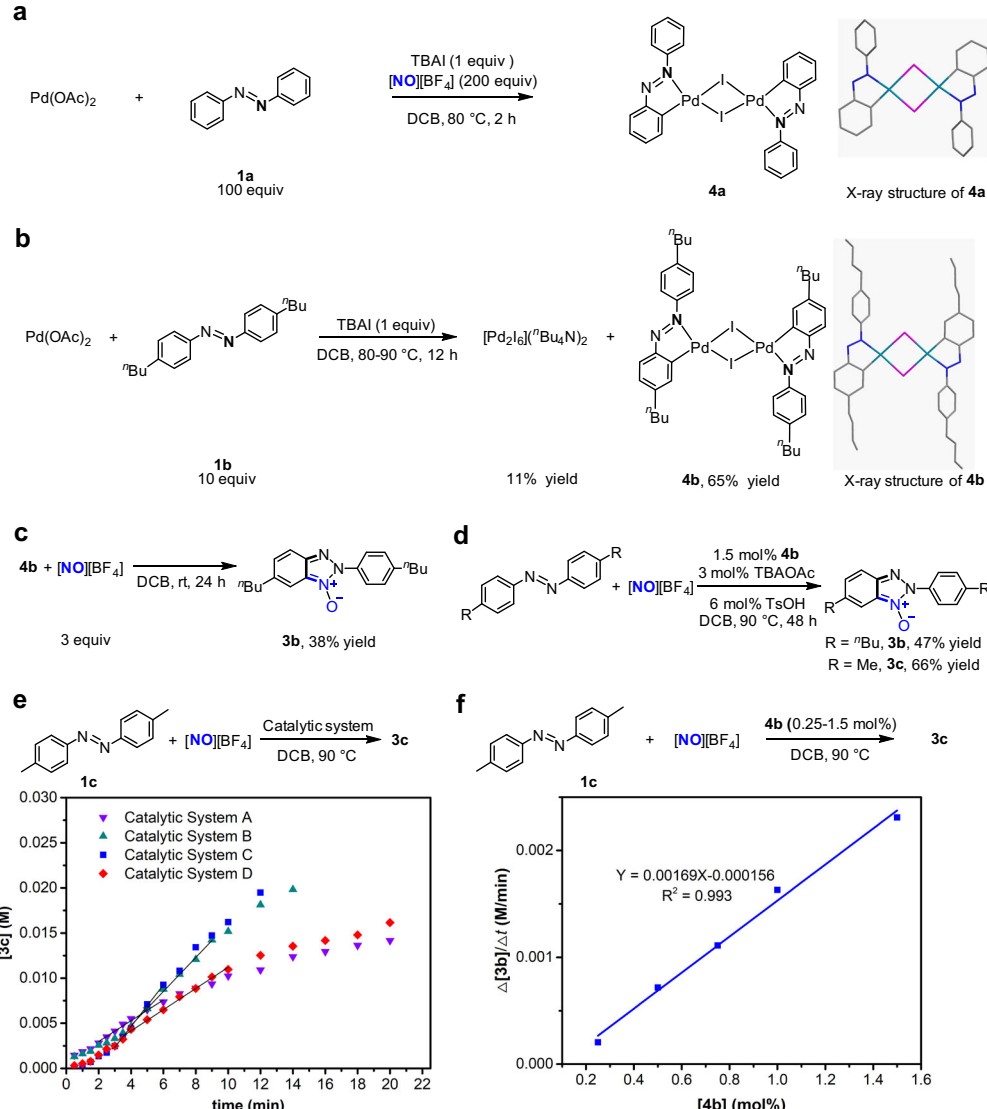

**Fig. 2 Detection, preparation and reactivity of iodide-bridged binuclear palladium complex bearing cyclopalladated azobenzene ligand. a** Isolation of iodide-bridged binuclear palladium complex bearing cyclopalladated azobenzene ligand (**4a**) from the reaction under conditions mimicking catalysis process. **b** Preparation of iodide-bridged binuclear palladium complex bearing cyclopalladated 4,4′-di-*n*-butyl-azobenzene ligand **4b** from self-assembly of 4,4′-di-*n*-butyl-azobenzene ligand and Pd(OAc)$_2$. **c** Stoichiometric reaction of **4b** with [NO][BF$_4$] at room temperature to generate 2*H*-benzotriazole *N*-oxide. **d** **4b**-catalysed reaction of 4,4′-di-substituted azobenzenes with [NO][BF$_4$]. **e** Comparison of activity of catalyst systems by determining the initial rates on the basis of plot of product 3c concentration (M) versus time (min). The reaction of 4,4′-dimethyl-azobenzene with [NO][BF$_4$] was used a model, which was analysed by HPLC with 3,4-dichlorotoluene as the internal standard. Catalyst system A is 3 mol% Pd(OAc)$_2$ and 6 mol% TsOH; catalyst system B is 3 mol% Pd(OAc)$_2$, 3 mol% TBAI and 6 mol% TsOH; catalyst system C is 1.5 mol% **4b**; catalyst system D is 1.5 mol% **5b** and 6 mol% TsOH. **f** A first-order dependence of the initial rate Δ[3c]/Δt (M/minute) on the **4b** concentration [**4b**] (mol%) catalyst, using the reaction of 4,4′-dimethyl-azobenzene with [NO][BF$_4$] as a model.

slightly faster than the reaction catalysed by the combination of 3 mol% Pd(OAc)$_2$ and 3 mol% TBAI since the concentration of the binuclear palladium complex generated in situ from self-assembly of 3 mol% Pd(OAc)$_2$, 3 mol% TBAI and azobenzene is likely less than 1.5 mol%.

To confirm that **4b** and its congeners are the catalytically active intermediates lying on catalytic cycle, we need to preclude the possibility that **4b** and its congeners are the off-cycle precatalysts. The acetate-bridged binuclear palladium complex bearing a cyclopalladated 4,4′-di-*n*-butyl-azobenzene ligand on each palladium centre (**5b**)[48] was observed to react with 2 equivalents of TBAI to lead to replacement of acetate bridge by iodide bridge and afford **4b** in 74% yield (Fig. 3a). The conversion of **5b** to **4b** in turn implicated that the iodide-bridged binuclear palladium

complexes were stable against the Pd–I bond cleavage caused through the ligand substitution, which is consistent with the fact that **4b** was generated from the reaction of Pd(OAc)$_2$ with excessive coordinating azobenzenes. Moreover, treatment of 0.1 mmol of **4b** with 3 equivalents of [NO][BF$_4$] and 6 equivalents of 4,4′-di-*n*-butyl-azobenzene in 5 mL DCB at 90 °C for 4 h gave rise to formation of 2*H*-benzotriazole *N*-oxide derivative **3b** in 133% yield relative to **4b** with 21% of **4b** recovered (Fig. 3b), which supported that **4b** could be re-generated during catalysis and that [NO][BF$_4$] did not fragment this binuclear palladium complex via oxidation of bridging iodide. [Pd$_2$I$_6$](*n*-Bu$_4$N)$_2$ (1.5 mol%), a side-product in the preparation of **4b** (Fig. 2b), was observed to catalyse the reaction of **1a** with [NO][BF$_4$] to afford **3a** in 82% yield (Fig. 3c). However, in the stoichiometric reaction of [Pd$_2$I$_6$]

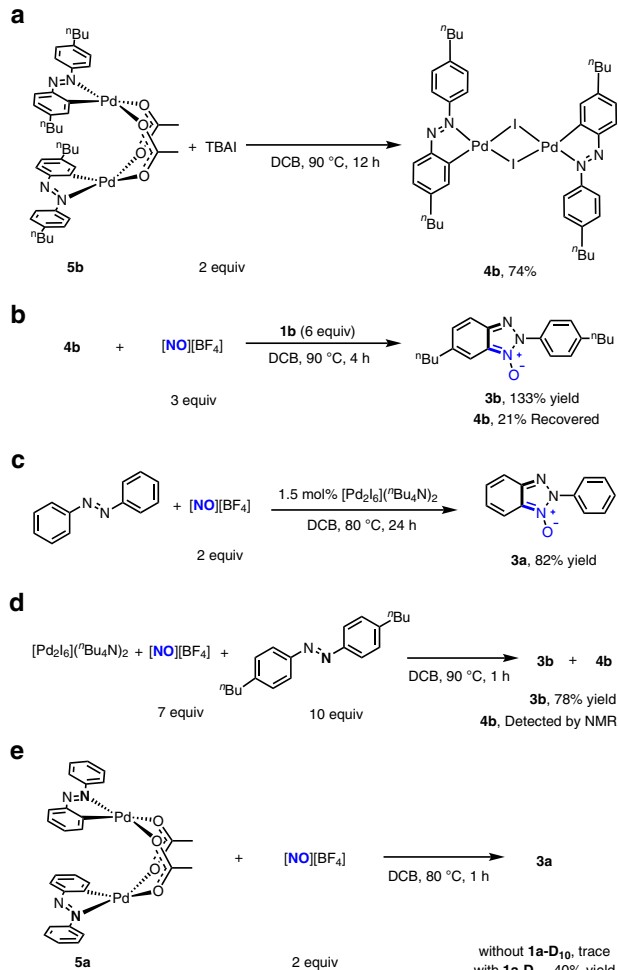

**Fig. 3 Reactivity of the palladium complexes related to 4b and their conversions to 4b. a** The reaction of acetate-bridged binuclear palladium complex bearing cyclopalladated 4,4′-di-n-butyl-azobenzene ligand (**5b**) with 2 equivalents of TBAI to generate **4b**. **b** Stoichiometric reaction of **4b** with 3 equivalents of [NO][BF₄] and 6 equivalents of 4,4′-di-n-butyl-azobenzene. The reaction formed **3b** in 133% yield relative to **4b** with 21% of **4b** recovered. **c** Catalytic reaction of azobenzene with [NO][BF₄] using 1.5 mol% [Pd₂I₆](n-Bu₄N)₂ as a catalyst. **d** Stoichiometric reaction of [Pd₂I₆](n-Bu₄N)₂ with 7 equivalents of [NO][BF₄] and 10 equivalents of 4,4′-di-n-butyl-azobenzene. The reaction formed **3b** and **4b**. **e** The reaction of acetate-bridged binuclear palladium complex bearing cyclopalladated azobenzene ligand **5a** with 3 equivalents of [NO][BF₄] with or without fully deuterated azobenzene (**1a-D₁₀**).

(n-Bu₄N)₂ with 7 equivalents of [NO][BF₄] and 10 equivalents of **1b** (Fig. 3d), **4b** was detected with **3b** formed, indicating that the catalytic behaviour of [Pd₂I₆](n-Bu₄N)₂ likely originated from its in situ conversion to the congener of **4b** via oxidation of terminal iodide ligand by [NO][BF₄][49]. The experimentally demonstrated stability of **4b** relative to other Pd species under the conditions similar to those for **4b**-catalysis supported that the robust [Pd₂I₂] core structure of **4b** retained integrity in the **4b**-catalysed reactions.

As shown in Fig. 2e, the reaction catalysed by the combination of 3 mol% Pd(OAc)₂ and 3 mol% TBAI was two times as fast as the reaction catalysed by 3 mol% Pd(OAc)₂, revealing that **4a** as a catalyst is more active than Pd(OAc)₂. 1.5 mol% **5b** provided the similar initial rate to 3 mol% Pd(OAc)₂, supporting the reaction catalyzed by Pd(OAc)₂ proceeds via cyclopalladation of azobenzenes to form analogues of **5b**. In line with this surmise, an

acetate-bridged binuclear palladium complex bearing cyclopalladated unsubstituted azobenzene ligand (**5a**)[48], an analogue of **5b**, reacted with [NO][BF₄], in the presence of one equivalent of deuterated azobenzene (**1a**-D₁₀) at 80 °C for 1 h, to produce the expected **3a** in 40% yield (Fig. 3e). In contrast, the stoichiometric reaction of **5a** with [NO][BF₄] gave none of **3a** in the absence of **1a**-D₁₀, which may implicate the reason why the reaction of **1a** with [NO][BF₄] catalyzed by 10 mol% Pd(OAc)₂ alone did not go to completion.

**Computational studies on the origin of the high activity of iodide-bridged binuclear palladium catalyst.** The density functional theory (DFT) calculations on the mechanism for the nitrosylation/annulation reaction of azobenzene with [NO][BF₄] catalysed by **4a**. For comparison, DFT studies of this process with Pd(OAc)₂ alone as the precatalyst were also carried out, in which acetate-bridged binuclear palladium complex **5a** is assumed to be the catalyst generated in situ on the basis of aforementioned investigations.

Figure 4a presents the computed reaction pathways with catalyst **4a** (blue line) and catalyst **5a** (red line). These two pathways comprise similar elemental step. Reactions begin with oxidative addition of nitrosonium to one of two palladium atoms in **4a** and **5a** to produce higher oxidation state nitrosyl-palladium intermediates (**LM1′** and **LM1**) (see Supplementary Fig. 16), which are followed by reductive elimination step to construct C–N bond and form six-membered chelation ring complexes (**LM2′** and **LM2**). Then, assisted by the coordination of azobenzene ligand to palladium centre, dechelation of such six-membered chelation rings via **TS2′** and **TS2**, and subsequent N–N bond formation for annulation proceed to release the target product 2H-benzotriazole N-oxide, respectively. Finally, with azobenzene as a base, azobenzene coordinating to palladium centre undergoes C–H palladation via a concerted metalation–deprotonation pathway to re-generate catalysts (**4a** and **5a**) from the corresponding σ-complexes **LM4′** and **LM4**. In the pathway of **4a**, redox steps involve the change of the formal oxidation state of only one palladium atom from II to IV[18], while in the pathway of **5a**, both palladium atoms synergistically participate in redox steps via switch between Pd(II)-Pd(II) and Pd (III)-Pd(III) oxidation states[17–19]. The rate-determining step in the pathway of **4a** is the C–H activation step with the activation barrier of 24.8 kcal/mol, in accordance with the experimentally observed intermolecular kinetic isotope effect (KIE) value of 4.4 (see Supplementary Fig. 14). In contrast, the step for dechelation of the palladium intermediate containing six-membered chelating ring is the rate-determining step in the pathway of **5a** with the activation barrier of 27.5 kcal/mol. The lower activation barrier in the pathway of **4a** is consistent with the observation that the **4b**-catalysed reaction is faster than **5b**-catalysed reaction (Fig. 2e).

Figure 4b shows the calculated structures of two palladium intermediates containing six-membered chelating rings (**LM2** and **LM2′**). As reflected by bond lengths of the structure of **LM2′**, phenyl part of cyclopalladated azobenzene weakens the trans Pd2–I2 bond due to its strong trans influence, and therefore strengthens the Pd1–I2 bond, which enhances the trans influence of I2 on Pd1–N1 bond to labilize Pd1–N1 bond for dechelation. As a result, the calculated bond length for Pd1–N1 bond of iodide-bridged binuclear palladium species is longer by 0.045 Å than the corresponding bond of acetate-bridged species. In line with the bond lengths, the calculated Pd1–N1 bond strength in **4a**-derived **LM2′** is weaker than that of the **5a**-derived **LM2** (Fig. 4c). The further calculations (Fig. 4b) on the dechelation step without aid of azobenzene illustrate that the dechelation/N–N bond formation step of **5a**-derived **LM2** is infeasible in

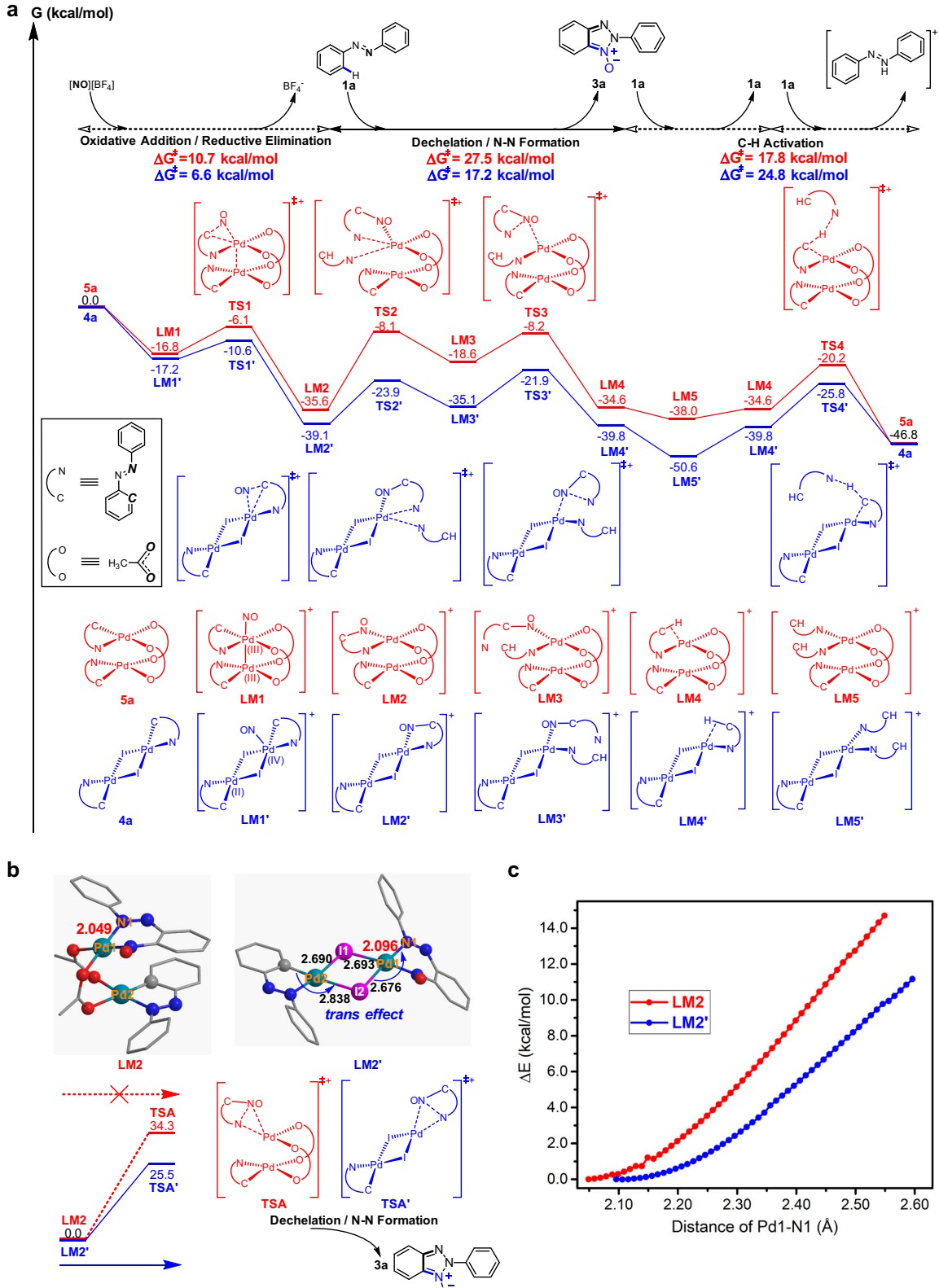

**Fig. 4 Computational investigation on the reaction mechanism. a** Free energy profiles for reaction catalyzed by **4a** (blue line) and **5a** (red line). **b** Optimized structures (bond length in Å) of iodide-bridged (**LM2′**) and acetate-bridged (**LM2**) binuclear palladium intermediates bearing six-membered chelating rings, and activation barriers for the dechelation/N-N bond formation steps from **LM2′** via **TSA′** transition state and **LM2** via **TSA** transition state in the absence of azobenzene. **c** Dissociation curves along Pd1–N1 bond for **LM2** (red line) and **LM2′** (blue line).

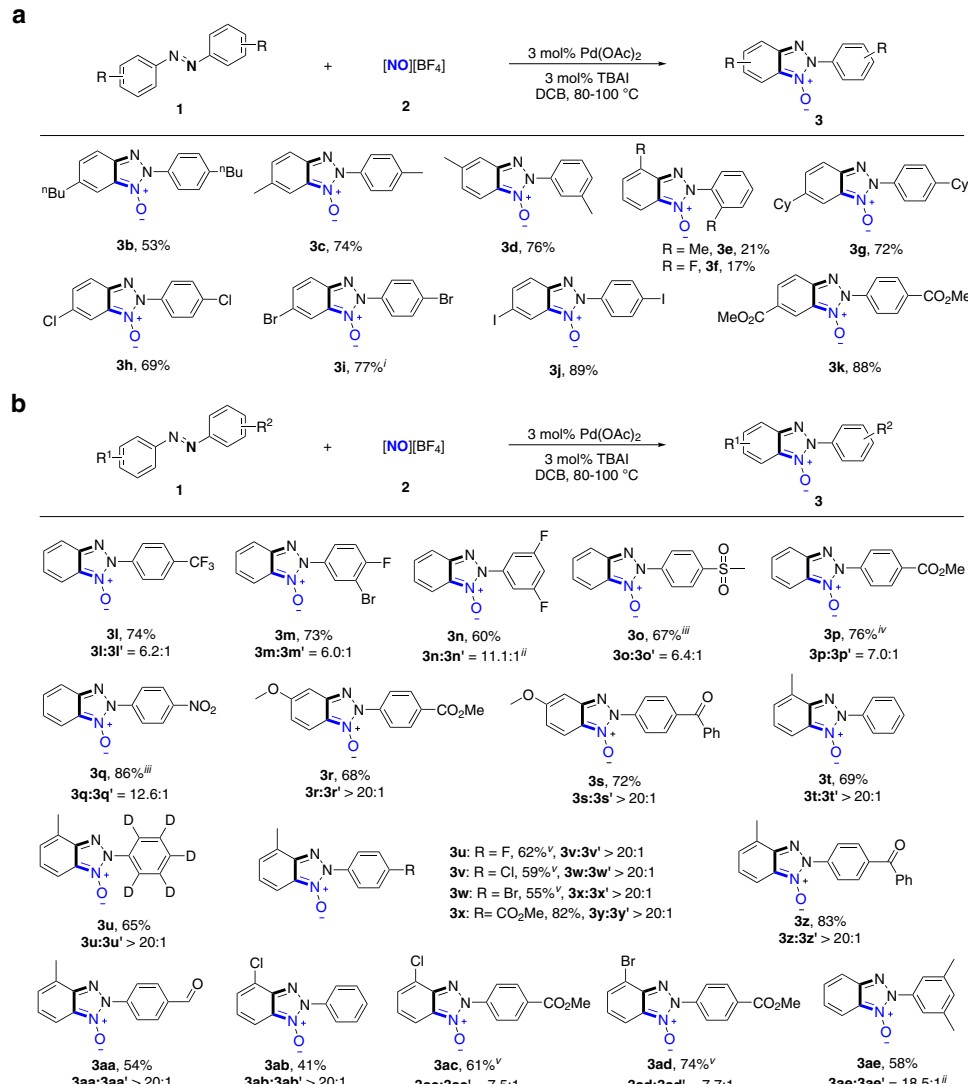

**Fig. 5 Substrate scope of Pd-catalyzed C–H nitrosylation/annulation reaction of azobenzenes with [NO][BF₄]. a** Substrate scope with respect to symmetrical azoarenes. **b** Substrate scope with respect to symmetrical azoarenes. Reaction conditions: azobenzene (0.2 mmol), [NO][BF₄] (3 equiv), Pd (OAc)₂ (3 mol%), TBAI (3 mol%), TsOH (6 mol%), DCB (1.5 mL), 90 °C, 48 h. Yields are isolated yields. [i]The reaction was run at 100 °C for 24 h. [ii]The ratio of product isomers was determined by NMR. [iii]The reaction was run in nitrobenzene (1.5 mL) at 100 °C for 24 h. [iv]0.6 mmol scale reaction was run at 80 °C for 24 h, with [NO][BF₄] (2 equiv), Pd(OAc)₂ (0.5 mol%), TBAI (0.5 mol%). [v]The reaction was run in DCB (1.0 mL) and nitrobenzene (0.5 mL) at 90 °C for 48 h.

kinetics with a high free energy barrier of 34.3 kcal/mol (**TSA**), while the same step of **4a**-derived **LM2′** only needs to experience a lower barrier of 25.5 kcal/mol (**TSA′**). Such a conclusion is supported by the experimental observations that in the absence of free deuterated azobenzene, the reaction of **5a** with [NO][BF₄] at 80 °C did not produce **3a** (Fig. 3e), while without any free azobenzene, **4b** reacted with [NO][BF₄] at room temperature to give the expected product (Fig. 2c), and that the 10 mol% Pd (OAc)₂-catalysed reaction of **1a** with [NO][BF₄] did not go completion (entry 2, Table 1). As such, the *trans* effect-relay through bridging iodide in the **4a**-derived binuclear palladium intermediate make the dechelation step easier and accelerates the reaction.

**Substrate scope**. This Pd-catalyzed C–H nitrosylation/annulation reaction of azobenzenes with [NO][BF₄] is quite general. To achieve the satisfied yields, some of reactions were conducted at

90 °C for 48 h with 3 mol% Pd(OAc)₂, 3 mol% TBAI and 6 mol% TsOH. In some cases, the combination of 0.5 mol% Pd(OAc)₂ and 0.5 mol% TBAI afforded good yields. As shown in Fig. 5a, a spectrum of 4,4′-di-substituted symmetrical azoarenes bearing *n*-butyl, methyl, cyclohexyl, chloro, bromo, iodo and ester substituents could participate in the reaction to generate the corresponding products in good-to-excellent yields (**3b-k**) with the exception of **3e** and **3f**. The low yield of **3e** and **3f** may result from the steric hindrance from *ortho*-substitutes, which impeded the ligation of azobenzene nitrogen atoms to palladium centre.

As shown in Fig. 5b, electronically unsymmetric azobenzenes containing an array of functional groups preferentially underwent reactions at more electron-rich benzene moiety with good-to-excellent yields obtained (**3l–3s**). For unsymmetric azobenzenes containing *ortho*-substituted benzene moieties, *ortho*-substituents led reactions to selectively occur at *ortho*-substituted benzene rings in good yields (**3t–3z**, **3aa–3ae**). *Ortho*-chloro group, an electron-withdrawing group, favoured reaction at chloro-containing

benzene moiety (**3ab**), which illustrated that the steric effect overrode the electronic effect in the control of reaction selectivity. The regioselectivity controlled by steric factor was also observed in 3,5-dimethyl azobenzene that favoured the reaction at less sterically hindered moiety with 18.5:1 selectivity (**3ae**).

In summary, an iodide-bridged binuclear palladium complex has been identified as an efficient catalyst to enhance both yield and rate of the nitrosylation/annulation reaction of azobenzenes with [NO][BF$_4$]. The good performance of this binuclear catalyst arises from the bimetallic cooperation by which the strongly σ-donating η$^1$ phenyl ligand on the "spectator" metal centre exerts a *trans* effect on the chelating product fragment on the catalytic metal centre through a bridging iodide ligand, and facilitates product release to re-generate catalytically active species. The *trans* effect-relay through bridging ligand within a binuclear complex represents a new bimetallic cooperation mode for catalysis and opens an avenue to design and develop multinuclear catalysts, especially for syntheses of the chelating products that often impede re-generation of active metal catalysts and therefore retard catalytic cycles.

## Methods

**General procedure for the reaction of azobenzene with [NO][BF$_4$]**. In a glove box, a 25 mL of the Schlenk tube equipped with a stir bar was charged with Pd (OAc)$_2$ (3 mol%, 0.0014 g), TBAI (3 mol%, 0.0022 g), p-toluenesulfonic acid (6 mol%, 0.0021 g), azobenzene (0.2 mmol), [NO][BF$_4$] (0.6 mmol, 0.0701 g) and DCB (1.5 mL). The tube was sealed and removed out of the glove box. The reaction mixture was stirred at 90 °C for 48 h. Upon completion, the reaction mixture was diluted with 10 mL of ethyl acetate, filtered through a pad of silica gel, followed by washing the pad of the silica gel with the ethyl acetate (20 mL). The filtrate was concentrated under reduced pressure. The residue was then purified by chromatography on silica gel to provide the corresponding product.

**Optimization studies**. See Supplementary Methods and Supplementary Table 1.

**Identification of iodide-bridged binuclear palladium complex as a catalytically active species**. See Supplementary Methods for details.

**Kinetic experiments**. See Supplementary Methods, Supplementary Tables 2–5 and Supplementary Figs. 1–5.

**Determination of the order in 4b**. See Supplementary Methods, Supplementary Tables 6–10 and Supplementary Figs. 6–11.

**KIE experiments**. See Supplementary Methods, Supplementary Table 11 and Supplementary Figs. 12–14.

**Typical procedure for the preparation of substrates**. See Supplementary Methods for details.

**Computational studies**. See Supplementary Methods, Supplementary Eq. 1 and Supplementary Figs. 15, 16. DFT computed Cartesian coordinates of important structures are provided in Supplementary Data 1.

**Crystallography**. The CIF files for compounds **3a**, **4a**, **4b** and **5b** are available in Supplementary Data 2–5, respectively. The check CIF/PLATON report file is available in Supplementary Data 6. Crystal data and structure refinement for compounds **3a**, **4a**, **4b** and **5b** are shown in Supplementary Tables 12–15.

**NMR spectra**. $^1$H NMR$^{13}$, C NMR and $^{19}$F NMR spectra of products, **4b** and **5b**. See Supplementary Figs. 17–103.

## Data availability

All data generated and analysed during this study are included in the article and its Supplementary Information file, or from the corresponding authors on request. Crystallographic data have been deposited at the Cambridge Crystallographic Data Centre (CCDC) as CCDC 1428672 (**3a**), CCDC 1858251(**4a**), CCDC 1858251(**4b**) and CCDC 1858266 (**5b**), and can be obtained free of charge from the CCDC via http://www.ccdc.cam.ac.uk/structures.

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

## Acknowledgements

This work was supported by the National Key Research and Development Program of China (2017YFA0206801), National Natural Science Foundation of China Grants (21602221, 21931011, 21431008, 21973094), the Strategic Priority Research Program of the Chinese Academy of Sciences (XDB20000000), the Key Research Program of Frontier Sciences of the Chinese Academy of Sciences (QYZDJ-SSW-SLH024) and the Natural Science Foundation of Fujian Province (2019J01131).

## Author contributions
Y.Z. and W.S. conceived and designed the project. Y.Z., X.Z. and X.D. performed experiments and analysed the data. Z.-N.C. and W.Z. performed computational studies. W.S., Y.Z., Z.-N.C. and W.Z. wrote and revised the manuscript.

## Competing interests
The authors declare no competing interests
