## [Peer Review File · Communications Chemistry]

Reviewers' comments:

Reviewer #1 (Remarks to the Author):

Iodide-enhanced palladium catalysis via formation of iodide-bridged 1 binuclear palladium complex
Su et al report a μ_2 -iodide-bridged binuclear palladium catalyst which accelerates the C–H nitrosation/annulation reaction and significantly enhances its yield compared with palladium acetate catalyst. This reaction bears n-butyl, methyl, cyclohexyl, chloro, bromo, iodo and ester substituents to generate the corresponding products in good to excellent yields. It is clear that obtained intermediate 4b reacts with [NO][BF₄] to generate the desired product. This mechanism indicated a better route to form the 2H-benzotriazole heterocyclic N-oxides than previous work.

Question 1: The hydrogen spectrum of compound 3p' should be made more pure and its low peaks could not exhibit more information for other scientific research workers.

Question 2: How about the substrates with double bond (alkenes and alkynes) and forming the corresponding products?

Question 3: Product 3e shows low yield (21%) because of the influence of ortho-Methyl group, have you tried other function groups on ortho-position and got the target compound?

Question 4: Table 1, 1 mol% TBAI is the same yield of product as 2 mol% TsOH in optimization of the reaction, how about other protonic acids as additive instead?

Question 5: 2H-benzotriazole heterocyclic N-oxides are the privileged structural motif in biologically active compounds, pharmaceuticals, and functional materials. Therefore, have you tried to test the corresponding biological activity and find a good result?

Reviewer #2 (Remarks to the Author):

The authors report on the discovery of iodide-bridged binuclear palladium complex-accelerated dechelation in the C-H nitrosylation/cyclization reaction of azobenzene and [NO][BF₄]. The work is significant because it provides a new bimetallic cooperation, in which trans-ligand accelerating effect was achieved through bridging iodide ligand. In addition, this method provides an alternative to facilitate difficult reductive eliminations. The detailed study on isolation, identification, function of catalyst complexes, kinetic experiments, as well as computational studies well supports the proposed trans-effect relay by iodide bridge. Reasonable functional group compatibility is demonstrated and products are obtained in moderate to good yields. Reasonable functional group compatibility is demonstrated and products are obtained in moderate to good yields with good regioselectivity. The manuscript was carefully and concisely written and provides a brief but thorough discussion of the most relevant literature. An adequate level of detail is provided in the experimental procedures and an appropriate level of analytical data including NMR spectra and X-ray are provided. The manuscript is recommended for publication with some minor concerns listed below.

1. The authors demonstrated "strongly σ -donating phenyl ligand" in the introduction and conclusion, but no evidence in their study shows phenyl acts as strongly σ -donating ligand. Either some comparisons of electronic effect of phenyl may be required to support this conclusion, or delete strongly σ -donating.
2. On page 1, abstract: "through the μ_2 -ligands with an absence of metal-metal bond" should be "through μ_2 -ligands without a metal-metal bond".

Point-to-point Responses to Comments of Reviewers

Manuscript ID: COMMSCHEM-19-0313-T (Yuanfei Zhang, Zhe-Ning Chen, Xiaofeng Zhang, Xi Deng, Wei Zhuang and Weiping Su)

Response to the comments of Reviewer #1.

General comments of Reviewer #1. Iodide-enhanced palladium catalysis via formation of iodide-bridged binuclear palladium complex by Su et al report a μ_2 -iodide-bridged binuclear palladium catalyst which accelerates the C–H nitrosation/annulation reaction and significantly enhances its yield compared with palladium acetate catalyst. This reaction bears n-butyl, methyl, cyclohexyl, chloro, bromo, iodo and ester substituents to generate the corresponding products in good to excellent yields. It is clear that obtained intermediate 4b reacts with [NO][BF₄] to generate the desired product. This mechanism indicated a better route to form the 2H-benzotriazole heterocyclic N-oxides than previous work.

Reply: We thank the reviewer for his/her comments that are very helpful for improvement of our manuscript.

Question 1: The hydrogen spectrum of compound 3p' should be made more pure and its low peaks could not exhibit more information for other scientific research workers.

Reply: We thank the reviewer for this suggestion. In order to get a more clear hydrogen spectrum of compound 3p', we did four parallel reactions using the same conditions as described in the original Supplementary Information. The newly obtained ¹H NMR of 3p' as well as ¹³C NMR are more clear than previous and could exhibit the needed information of 3p'. Duo to the addition of 3f to Figure 5 in Manuscript and Supplementary Information, we have changed the ¹H NMR and ¹³C NMR of 3p' to 3q' in the revised Supplementary Information and listed as follow:

Question 2: How about the substrates with double bond (alkenes and alkynes) and forming the corresponding products?

Reply: In response to this comment, we have prepared azobenzenes with double and triple bond listed as follow. Unfortunately, the substrates decomposed under the standard reaction conditions, and therefore failed to deliver the target products. The decomposition of these substrates may stem from the high redox potential of $[\text{NO}][\text{BF}_4]$. The searching for mild nitrosation source is ongoing in our group.

Question 3: Product 3e shows low yield (21%) because of the influence of ortho-Methyl group, have you tried other function groups on ortho-position and got the target compound?

Reply: We thank the reviewer for this suggestion. We have synthesized (E)-1,2-bis(2-methoxyphenyl)diazene and (E)-1,2-bis(2-fluorophenyl)diazene to investigate the influence of *ortho* substituents (listed below). Azobenzene with OMe substitutes on *ortho*-position failed to participate in the reaction and the desired product was not obtained. While the target product was isolated in 17% yield when (E)-1,2-bis(2-fluorophenyl)diazene was employed as substrate. Those results indicated that the reactions of symmetric azobenzenes with *ortho* substituents are retarded with the established reaction conditions, which may result from the *ortho* substituents impeded the ligation of azobenzene nitrogen atoms to palladium center. The product 4-fluoro-2-(2-fluorophenyl)-2*H*-benzo[*d*][1,2,3]triazole 1-oxide was added to Figure 5 as 3f in the revised manuscript and the ^1H NMR, ^{13}C NMR and ^{19}F NMR of 3f was added to the Supplementary Information as Supplementary Figure 27, 28 and 29.

Question 4: Table 1, 1 mol% TBAI is the same yield of product as 2 mol% TsOH in optimization of the reaction, how about other protonic acids as additive instead?

Reply: To elucidate the effect of other protonic acids on the reaction yield, HOAc, PivOH, benzoic acid and 4-nitrobenzoic acid were respectively added as additive. The starting azobenzene 1a was consumed completely in all the four reactions, and the isolated yields summarized below are all around 90%. In fact, full conversion of 1a was also observed when TsOH was utilized as additive. Those results were added to Supplementary Table 1 in the revised Supplementary Information. "Other protonic acids did not show beneficial effect on the reaction yield as well (see Supplementary Table 1)" was added to the revised manuscript.

Question 5: 2H-benzotriazole heterocyclic N-oxides are the privileged structural motif in biologically active compounds, pharmaceuticals, and functional materials. Therefore, have you tried to test the corresponding biological activity and find a good result?

Reply: We agree that evaluation of the biological activity is valuable. However, the biological activity of the obtained 2H-benzotriazole heterocyclic N-oxides has not been tested at this stage. We thank the reviewer for this reminding and will concern on the biologically relevant properties in the future.

Response to the comments of Reviewer #2.

General comments of Reviewer #2: The authors report on the discovery of iodide-bridged binuclear palladium complex-accelerated dechelation in the C-H nitrosylation/cyclyzation

reaction of azobenzene and [NO][BF₄]. The work is significant because it provides a new bimetallic cooperation, in which trans-ligand accelerating effect was achieved through bridging iodide ligand. In addition, this method provides an alternative to facilitate difficult reductive eliminations. The detailed study on isolation, identification, function of catalyst complexes, kinetic experiments, as well as computational studies well supports the proposed trans-effect relay by iodide bridge. Reasonable functional group compatibility is demonstrated and products are obtained in moderate to good yields. Reasonable functional group compatibility is demonstrated and products are obtained in moderate to good yields with good regioselectivity. The manuscript was carefully and concisely written and provides a brief but thorough discussion of the most relevant literature. An adequate level of detail is provided in the experimental procedures and an appropriate level of analytical data including NMR spectra and X-ray are provided. The manuscript is recommended for publication with some minor concerns listed below.

Reply: We thank the reviewer for his/her comments that are very helpful for improvement of our manuscript.

Question 1. The authors demonstrated “strongly σ -donating phenyl ligand” in the introduction and conclusion, but no evidence in their study shows phenyl acts as strongly σ -donating ligand. Either some comparisons of electronic effect of phenyl may be required to support this conclusion, or delete strongly σ -donating.

Reply: We thank the reviewer for this reminding. We have realized that the phenyl ligand could not be simply defined as “strongly σ -donating” since there are many possible coordination modes for phenyl ligand. According to the definition in organometallic chemistry, only the phenyl ligand with η^1 coordination mode can be considered as “strongly σ -donating” and therefore is able to exert a *trans* effect (see Coe, B. J. & Glenwright, S. J. Trans-effects in octahedral transition metal complexes. *Coord. Chem. Rev.* 203, 5-80 (2000)). In response to this comment, we have changed “strongly σ -donating phenyl ligand” to “strongly σ -donating η^1 phenyl ligand” in the revised manuscript, and a new reference of *Coord. Chem. Rev.* 203, 5-80 (2000) has been added.

2. On page 1, abstract: “through the μ_2 -ligands with an absence of metal-metal bond” should be “through μ_2 -ligands without a metal-metal bond”.

Reply: We thank the reviewer for this reminding. We have revised this sentence according to this comment.

REVIEWERS' COMMENTS:

Reviewer #1 (Remarks to the Author):

accept

Reviewer #2 (Remarks to the Author):

All concerned questions have been addressed, the manuscript is recommended to publish.

Manuscript ID: COMMSCHEM-19-0313A (Yuanfei Zhang, Zhe-Ning Chen, Xiaofeng Zhang, Xi Deng, Wei Zhuang and Weiping Su)

Reviewer #1 (Remarks to the Author):

Comment 1. accept

Response: We thank the reviewer for his/her time and efforts in reading and making very important comments to the manuscript. His comments indeed greatly helped us to improve our manuscript.

Reviewer #2 (Remarks to the Author):

Comment 1. All concerned questions have been addressed, the manuscript is recommended to publish.

Response: We thank the reviewer for the kind recommendation to publish this work.